# Examining intentions to take iron supplements to inform a behavioral intervention: The Reduction in Anemia through Normative Innovations (RANI) project

Erica Sedlander[1], Michael W. Long[1]*, Jeffrey B. Bingenheimer[1], Rajiv N. Rimal[1,2]

1 Department of Prevention and Community Health, Milken Institute School of Public Health, The George Washington University, Washington, DC, United States of America, 2 Department of Health Behavior and Society, Johns Hopkins University, Baltimore, Maryland, United States of America

* michael_long@gwu.edu

## Abstract

### Background

More than half of women of reproductive age in India have anemia. Over the last decade, India has made some progress towards reducing anemia in pregnant women, but non-pregnant women, who make up the largest sub group of people with anemia, are largely disregarded.

### Objectives

The objective of this paper is to examine intentions to take iron supplements and factors associated with intentions to inform a social norms-based behavioral intervention to increase uptake of iron supplements and reduce anemia in Odisha, India.

### Methods

We collected data from 3,914 randomly sampled non-pregnant women of reproductive age in 81 villages. We conducted a survey and took hemocue (anemia level) readings from each participant. We analyzed data using linear regression models beginning with demographics and social norms and adding other factors such as self-efficacy to take iron supplements, anemia risk perception, and knowledge about anemia in a subsequent model.

### Results

63% of women in our sample were anemic but less than 5% knew they were anemic. Despite national guidelines that all women of reproductive age should take weekly iron supplements to prevent anemia, less than 3% of women in our sample were currently taking them. While actual use was low, intentions were rather high. On a five point Likert scale where higher numbers meant more intentions to take supplements, average intentions were above the midpoint ($M = 3.48$, $SD = 1.27$) and intentions and iron supplement use were significantly correlated ($r = .10$, $p < .001$). Both injunctive norms and collective norms were

**Data Availability Statement:** The dataset for this manuscript has been made available in the George Washington University repository, GW

ScholarSpace (https://scholarspace.library.gwu.
edu/work/gx41mj67p).

**Funding:** This work was supported by a grant from
The Bill and Melinda Gates Foundation
(OPP1182519) to the George Washington
University, Rajiv N. Rimal, principal investigator.
The funders had no role in study design, data
collection and analysis, decision to publish, or
preparation of the manuscript.

**Competing interests:** The authors have declared
that no competing interests exist.

associated with intentions to take iron supplements but descriptive norms were not. Other
significant factors included age, breastfeeding, knowledge, self-efficacy, and outcome
expectations. The final model accounted for 74% of the variance in iron supplement
intentions.

## Conclusions

In this context, where the actual behavior is low but intentions to enact the behavior are
high, starting an intervention with injunctive norms messaging (expectations around the
behavior) and self-efficacy to enact the behavior is the step we recommend based on our
results. As an intervention unfolds and iron supplement use increases, descriptive norms
messaging (that people are indeed taking iron supplements) may add value.

## Background

More than a third of women of reproductive age worldwide and more than half of women in
India have anemia [1]. Given this high prevalence, a 50% reduction in anemia among all
women of reproductive age is the the World Health Assembly's global target by the year 2025
[2]. The vast majority of women in India with anemia have mild-to-moderate anemia (defined
as hemoglobin levels between 8–11.9 g/dL for non-pregnant women), which affects both work
capacity and productivity [3]. Anemia during pregnancy can result in adverse health out-
comes, including maternal mortality and low birthweight [4]. While the intergenerational
nature of anemia via pregnancy is critical, women's health by itself, apart from her children or
her role as mother, is also important. Both pregnant and non-pregnant women should be sup-
ported to be healthy and anemia free. Although pregnant women have the highest prevalence
of anemia due to biological changes during pregnancy, non-pregnant women make up the
largest number of individuals with anemia [1].

The World Health Organization recommends regular iron and folic acid (IFA) supplemen-
tation to prevent anemia for all women of reproductive age [5]. High rates of anemia in India
and other parts of southeast Asia reflect underlying social determinants, such as poverty,
which results in diets low in iron, high levels of malaria, and other infectious diseases [6].
While food-based interventions to increase iron intake are promising, they are difficult to
measure, and they have limited evidence of effectiveness at scale [7,8]. Therefore, the Indian
government has been actively promoting and distributing free iron folic acid (IFA) for preg-
nant and lactating women since 1970 [9]. The government has also been distributing weekly
IFA to adolescent girls at school since 2000 [10,11]. Later in 2013, the government adopted a
more comprehensive reproductive life-cycle approach under the National Iron Plus Initiative
(NIPI), which recommends weekly IFA for all women of reproductive age, regardless of ane-
mia or pregnancy status [12]. Most recently, in 2018, the government rolled out the Intensified
National Iron Plus Initiative "Anemia Mukt Bharat," with the goal of increasing IFA uptake
among non-pregnant women [13]. Since 2013, the World Health Organization and national
guidelines state that all women of reproductive age should be taking weekly IFA. However, a
significant gap exists between what is stated in the guidelines versus what is available and
being promoted on the ground [14].

To design effective interventions to increase IFA use, it is critical to examine multiple fac-
tors associated with intentions to take IFA. However, extant research focuses on pregnant

women and adolescents, disregarding non-pregnant women of reproductive age. Prior qualitative work has elucidated *why* non-pregnant women were not taking IFA. In our formative research, we found low perceptions of anemia prevalence in the community, compared to national data, and lack of knowledge among both women of reproductive age and health providers that taking IFA is recommended for non-pregnant women [14]. To our knowledge, no studies have quantitatively examined which factors affect non-pregnant women's intentions to take IFA.

The prior research on pregnant women shows that adequate knowledge and positive beliefs about IFA are fundamental [15–17]. Qualitative research also shows that social norms pertaining to IFA consumption are critical factors in driving this behavior [14,18].

### Conceptual background on social norms

In the broader literature, including in the focus theory of normative conduct [19], behaviors are conceptualized as driven by two types of norms–descriptive norms, which speak to people's perceptions about the prevalence of a behavior, and injunctive norms, pressures people feel to conform. The theory posits that behaviors are driven by norms that are made focal at the time of action.

In the theory of planned behavior [20], social norms are conceptualized as people's beliefs about what others would want them to do combined with their motivation to comply. This theory's conceptualization of norms, which are closer to injunctive norms in the focus theory of normative conduct, does not accommodate the role of descriptive norms.

The theory of normative social behavior (TNSB) also includes descriptive and injunctive norms, and its recent conceptualization has also added the idea of collective norms [21], which refer to the proportion of individuals actually engaging in the focal behavior in one's social midst. The TNSB also articulates various factors that serve to amplify normative influences, including one's identification with the group, something also articulated in social identity theory [22,23].

For this study, we drew on the TNSB, which tries to uncover when, how, and which norms are influential. We focused on two key constructs from the TNSB, which itself was derived from the focus theory of normative conduct [19]: *descriptive norms* and *injunctive norms*. Theories on social norms state that perceived descriptive and injunctive norms may impact behavioral intention (intent to take IFA), which may in turn impact one's behavior (actually taking IFA) [24–26]. However, descriptive and injunctive norms may not always align and even if they do, given that they are malleable and socially communicated, they can change at different times [26,27]. In other words, perceptions of the prevalence of the behavior (descriptive norms) may be different from perceptions regarding others' expectations of the behavior (injunctive norms). In a context where the prevalence of a behavior is low (e.g., non-pregnant women do not take IFA) and perceived descriptive norms are accurate, a campaign that attempts to correct or align perceived descriptive norms with the reality could be counterproductive. In this case, attempting to change or bolster perceived injunctive norms (expectations that non-pregnant should be taking IFA) may be a more fruitful initial strategy for a social norms intervention. Therefore, it is critical to examine both and to design an intervention based on the community's current behaviors, perceived descriptive and injunctive norms, and readiness to change the behavior. To our knowledge, this is the first field study to use a social norms approach to increase IFA consumption.

### Objectives

1. Describe the current state of intentions to take iron folic acid supplements and behaviors among non-pregnant women in Odisha, India

2. Identify factors associated with intentions to take iron folic acid supplements to inform a social norms-based intervention to increase use.

## Methods

This study was approved by the George Washington University Institutional Review Board (IRB #180187) and Sigma Science and Research (#10031/IRB/D/18-20), an IRB located in New Delhi, India. The study was also reviewed and approved by Indian Council for Medical Research's (ICMR's) Health Ministry's Screening Committee (HMSC).

### Study setting

Odisha is a coastal state in eastern India where most households are situated in rural areas. Almost all heads of household are Hindu (95%), and 23% of households belong to the tribal culture [3]. Of the many castes and tribes within Odisha, those who belong to the scheduled tribe have been the most marginalized followed by scheduled caste [28]. The state is divided into 30 districts, one of which, Angul, where this study takes place, has a population of just over 1.2 million. In Angul, 70% of women are literate, compared to 87% of men. 22% of women marry before age 18 and almost half of all married women of reproductive age are using modern methods of family planning. 44% of non-pregnant reproductive-age women are anemic [3].

### Study design

Data for this study come from the Reductions in Anemia through Normative Innovations (RANI) project [29]. The RANI project will use a cluster randomized controlled trial (cRCT) to evaluate the ability of a norms-based IFA promotion intervention in Angul, Odisha, India. We selected two blocks (a block is analogous to a county in the United States) and then grouped contiguous villages in these blocks into clusters, which we then randomly assigned to either the treatment or control arm. We then segmented clusters by the proportion of minority populations (in India they are called scheduled castes and scheduled tribes).We then selected three clusters from each stratum for data collection (representing the clusters with the lowest, medium, and highest proportion of scheduled caste/schedule tribe in each strata) so that 15 clusters (which comprised 41 villages) from the treatment arm and 15 (comprising 40 villages) from the control arm were selected for data collection (81 villages total).

In this paper, we report results from the baseline assessment, which is a cross-sectional study from both the treatment and control clusters. Because we collected data before the intervention was implemented, in this paper we have not made distinctions between the two study arms.

### Participants

In each designated village, we first enumerated all households and then randomly selected households for data collection. From the selected home, one woman of reproductive age (between 15 and 49 years old) was chosen (randomly if more than one woman was eligible in the same home). This paper restricts the sample to those who were not pregnant at the time of survey (n = 3,914) because their intentions and behaviors around taking IFA are vastly different from pregnant women [14]. The RCT from which these data are drawn was powered to detect changes in hemoglobin levels. We did not conduct a post-hoc power analysis for the current study given the limitations of this approach [30], but note that this study has a large sample relative to the literature on behavioral intentions.

## Procedure

Local research assistants who participated in a two-week training obtained informed consent to participate in the study in the local language of Odiya. Participants under the age of 18 were required to obtain the written permission of one parent or legal guardian. Research assistants orally administered a one-on-one tablet-based survey to all participants. 93% of women who were eligible, consented to join the study; only 47 refused to participate and 271 were not home. Our data collection partners returned to each house up to three times to achieve such a high response rate.

## Inclusion criteria

Women were eligible for the study if they were between the ages of 15 and 49, spoke Odiya, lived in the data collection villages, and did not plan to move in the next year (as this was part of the baseline data collection for a longitudinal study).

## Measures

**Demographic descriptors.**   We asked respondents about their pregnancy status, their age, highest completed level of education, whether or not they owned a mobile phone (a proxy for socio-economic status), marital status, religion, number of children they had, and whether or not they were breastfeeding. Additionally, we asked if they were currently taking IFA, if they believed they were anemic, and if a doctor of healthcare provider ever told them they were anemic.

**Hemoglobin measurements.**   We obtained hemoglobin levels from all participants, through point-of-care hemoglobin tests, using a HemoCue photometer (in line with India's National Family Health Survey methodology). This instrument provides hemoglobin levels immediately and accurately via a simple finger prick [31].

**Intentions to take IFA.**   We asked respondents the extent to which they agreed with the following three statements: "You will take iron and folic acid once a week in the future, even if you are not pregnant," "If you are not pregnant you will take iron and folic acid every week even if your husband/male member in your community does not think it is a good idea," and "If you are not pregnant you will take iron and folic acid every week even if your mother-in-law/woman in your community does not think it is a good idea." All responses were recorded on a 5-point Likert scale and averaged into an index for intentions ($\alpha = 0.88$; $M = 3.48$, $SD = 1.27$).

**Descriptive norms.**   We operationalized descriptive norms as people's beliefs about others' behaviors (regularly taking IFA supplements). We assessed descriptive norms with one question: "What proportion of non-pregnant women in your community take IFA regularly?" Responses were recorded on a 5-point scale, where 0 corresponded to "none," 1 corresponded to "some," 2 corresponded to "about half," 3 corresponded to "most," and 4 corresponded to "all" ($M = 0.25$, $SD = 0.51$).

**Injunctive norms.**   We operationalized injunctive norms as the extent to which women believed their influential referent groups (women in the community, husbands, and mothers-in-law or mothers) expected them to take iron and folic acid supplements. Respondents were asked to assess the strength of perceived injunctive norms from three reference groups using the following three questions: "How many of the women in your community (hamlet or village) think you should take iron tablets, even when you are not pregnant," "Your mother-in-law (or mother, if unmarried) thinks you should take iron and folic acid tablets regularly, even if you are not pregnant," and "Your husband (or father, if not married) thinks you should take iron tablets regularly, even if you are not pregnant." We chose husbands and mothers-in-law

as the referent groups because of our formative work that highlighted their importance [14]. Responses were recorded on a 5-point Likert scale and averaged into a single measure for injunctive norms (α = .75; $M$ = 1.8, $SD$ = .89).

**Collective norms.** We operationalized collective norms as the prevalence of intentions to take iron and folic acid within a cluster, calculated by computing the non-self-mean of intentions within each village. This method of calculating collective norms by aggregating behaviors within a community or selected peer group has been used in other studies [21,32]. The difference here is that we used intentions rather than behavior to understand how we can improve intentions and subsequently IFA use for an intervention.

**Knowledge.** To create a knowledge index, we took the average response to seven questions that asked about IFA and anemia information. All responses were dichotomous (true/false or correct/incorrect) and we recoded the responses so higher scores = more accurate knowledge of IFA and anemia. We included both accurate statements such as "Eating dark green leafy vegetables prevents anemia" and inaccurate statements such as, "Anemia can be spread from one person to another through their saliva" to truly understand participants comprehension of anemia and IFA (α = .63; $M$ = .47, $SD$ = .18).

**Efficacy.** We measured self-efficacy by asking participants the extent to which they agreed with the statements "You can take iron and folic acid every week when you are not pregnant," "You believe that you could easily take iron and folic acid," "You can take iron and folic acid even if your husband/father does not want you to do so," and "You can take iron and folic acid even if your mother/mother-in-law does not want you to do so." All responses were recorded on a 5-point Likert scale (α = .81; $M$ = 3.8, $SD$ = 1.16).

**Outcome expectations.** We measured outcome expectations (beliefs around the benefits of iron supplements) with one question, "Taking iron batika (tablets) regularly will make you feel stronger." Responses were recorded on a 5-point Likert scale ($M$ = 4.3, $SD$ = .89).

**Perceived access.** We measured perceived access with one question, "Do you agree or disagree with the statement, It is easy for you (or someone) to get iron batika (tablets)?" Responses were recorded on a 5-point Likert scale ($M$ = 4.2, $SD$ = .99).

**Risk perception.** We operationalized risk perception as the perceived level of susceptibility of getting anemia for oneself and one's family. Pilot testing of measurement items indicated that it was difficult for our population to answer scaled questions in assessing risk (likely because it required making probability judgments), so we used a dichotomous measure, instead. Participants were asked, "do you believe that you will become anemic in the coming year?" and "do you think someone in your family will become anemic in the coming year?" The resulting index for risk perception ranged from 0 to 2 with zero representing "no," 1 representing "yes," and 2 representing "already anemic" (α = 0.87; $M$ = .57, $SD$ = 0.47).

## Statistical analysis

We conducted our analyses in four steps. First, we calculated descriptive statistics. We then performed bivariate Pearson's correlations and multivariable linear regressions analyses to identify factors associated with intentions to take IFA. The first model contained demographic variables and social norms variables (descriptive, injunctive, and collective norms). The second model included all variables in model 1 and knowledge, self-efficacy, outcome expectations, perceived access, and risk perception. We standardized all variables before adding them to the models. We used STATA, version 14 to conduct all analyses. To obtain a robust variance estimate that adjusts for within-cluster correlation, we used the Huber-White clustered standard errors command [33].

## Results

Description of the sample included in our study is shown in **Table 1**. Average age was 30 years old. Almost a fifth (18%) had no school, about a quarter (24%) completed up to class five, and more than half (52%) completed up to class 12. Almost the entire sample (99%) was Hindu.

**Table 1. Description of the sample.**

|  | (N = 3,914) |
|---|---|
| **Age** | **Mean = 30.58** |
|  | SD (8.83) |
|  | % |
| **Education** |  |
| No school | 18.6 |
| Completed up to class 5 | 24.7 |
| Completed up to class 12 | 52.7 |
| More than class 12 | 3.5 |
| **Religion** |  |
| Hindu | 99.8 |
| Christian | 0.1 |
| **Scheduled caste, scheduled tribe, other backwards caste or none** |  |
| Scheduled caste | 14.1 |
| Scheduled tribe | 28.0 |
| Other backward caste | 56.0 |
| None of them | 2.1 |
| **Marital status** |  |
| Single | 16.5 |
| Married | 79.2 |
| Divorced | 0.2 |
| Separated | 0.7 |
| Widowed | 3.4 |
| **Number of Children** |  |
| None | 22.7 |
| One or two | 55.0 |
| Three or Four | 20.1 |
| Five | 2.3 |
| **Currently Breastfeeding** |  |
| Yes | 21.8 |
| No | 78.1 |
| **Owns a Mobile Phone** |  |
| Yes | 48.1 |
| No | 51.9 |
| **Ever Taken Iron Folic Acid Supplements** |  |
| No, I have never taken it | 22.1 |
| Yes I took in the past, but not current | 75.0 |
| Yes, I am currently taking it | 2.9 |
| **Have you asked for iron supplements in the last 6 months?** |  |
| Yes | 5.6 |
| No | 94.3 |
| **Anemia Status (Based on Hemoglobin Level)** |  |
| Yes | 63.1 |

(*Continued*)

**Table 1.** (Continued)

| | (N = 3,914) |
|---|---|
| **Age** | Mean = 30.58 |
| | SD (8.83) |
| | % |
| No | 36.9 |
| **Anemia Status by category (Based on Hemoglobin Level)** | |
| Not anemic (hemoglobin >12) | 36.5 |
| Mild (hemoglobin 11–11.9) | 31.58 |
| Moderate (hemoglobin 8–10.9) | 26.4 |
| Severe (hemoglobin <8) | 1.9 |
| **Has a doctor or a nurse ever told you that you have anemia? (self-reported)** | |
| Yes | 15.8 |
| No | 84.2 |
| **Do you currently have anemia? (self-reported)** | |
| Yes | 5.0 |
| No | 56.5 |
| Don't know | 38.5 |

14% of the sample belonged to a scheduled caste, just over a quarter belonged to a scheduled tribe, and more than half (56%), belong to "other backward caste." Most women in the sample (79%) were married and 16% were single. Almost a quarter of the sample had no children (22%), more than half (55%) had one or two children, 20% had three or four, and 2% had five or more. 21% of the sample was currently breastfeeding. About half (48%) owned a mobile phone.

Only 2.9% of the sample was currently taking IFA. Approximately 75% (about the percentage of women who have children) stated that they had taken it in the past but are not currently taking it and 22% had never taken it. Only 5% of women had asked for IFA in the last six months. Despite this, based on hemocue tests, 63% of women in the sample tested positive for anemia (hemoglobin level <12 g/dL). Of those who fell below this cutoff hemoglobin level for anemia, the majority had mild to moderate anemia, and only 1.9% had severe anemia. Women's self-reported anemia ran counter to their actual anemia rates. Only 5% said that they currently had anemia, 38% reported that they did not know if they had it, and most, 56% said that they did not have it. Lastly, only 15% said that a nurse or doctor ever told them they had anemia.

Table 2 shows the correlations, which indicate that intentions to take IFA were significantly associated with all variables in the model, but the largest effects were self-efficacy to take IFA ($r = .84$, $p < .001$), outcome expectations ($r = .38$, $p < .001$), injunctive norms around IFA ($r = .40$, $p < .001$), and collective level intentions to take IFA ($r = .32$, $p < .001$). This table also illustrates the associations between all of the variables included in the model and shows that while most are significantly associated with one another, the associations are not multi-collinear. For example, collective norms was significantly associated with descriptive norms ($r = .14$, $p < .001$), injunctive norms ($r = .22$, $p < .001$), and self-efficacy ($r = .30$, $p < .001$). Similarly, descriptive and injunctive norms were significantly associated ($r = .33$, $p < .001$).

Linear regressions (Table 3) showed that in model 1, which included demographic and social norms variables, education was positively associated with intention to take IFA ($\beta = .05$, $p < 0.05$). Injunctive norms that non-pregnant women should be taking IFA ($\beta = .36$, $p < .001$) and collective norms (an aggregate of IFA intentions within one's village minus the respondent herself) ($\beta = .25$, $p < .001$) were also both positively associated with intentions. This model predicted 23% of the variance in intentions to take IFA.

**Table 2. Pearson correlations.**

|  | 1 | 2 | 3 | 4 | 5 | 6 | 7 | 8 | 9 |
|---|---|---|---|---|---|---|---|---|---|
| 1 Intentions to take IFA | 1.00 | | | | | | | | |
| 2 Descriptive norms | 0.13*** | 1.00 | | | | | | | |
| 3 Injunctive norms | 0.40*** | 0.33*** | 1.00 | | | | | | |
| 4 Knowledge | 0.13*** | 0.02 | 0.14*** | 1.00 | | | | | |
| 5 Self-efficacy | 0.84*** | 0.12*** | 0.34*** | 0.10*** | 1.00 | | | | |
| 6 Risk perception | 0.10*** | 0.07*** | 0.08*** | 0.03 | 0.11*** | 1.00 | | | |
| 7 Perceived access | 0.20*** | 0.11*** | 0.16*** | 0.03 | 0.21*** | -0.03 | 1.00 | | |
| 8 Outcome expectations | 0.38*** | 0.10*** | 0.22*** | 0.10*** | 0.38*** | 0.05** | 0.14*** | 1.00 | |
| 9. Collective norms | 0.32*** | 0.14*** | 0.22*** | 0.05** | 0.30*** | 0.13*** | 0.14*** | 0.15*** | 1.00 |

Notes: Iron folic acid supplement (IFA). Collective norms refer to collective norms around intentions to take iron folic acid supplements (IFA) and are calculated as the non-self mean of respondents within each village without the target person.

*p < .05

**p < .01

***p < .001.

In model 2, we added knowledge of IFA and anemia, self-efficacy to take IFA, outcome expectations, anemia risk perception, and perceived access to the model. (Self-efficacy to take IFA had the greatest association with intentions to take IFA ($\beta$ = .77, $p < .001$). Knowledge around anemia and IFA ($\beta$ = .02, $p < 0.05$) and outcome expectations ($\beta$ = .05, $p < .001$), and breastfeeding ($\beta$ = .01, $p < 0.05$) were also positively associated with intentions. Older age was negatively associated with intentions to take IFA ($\beta$ = -.00, $p < 0.01$). The strength of injunctive

**Table 3. Associations with intentions to take iron folic acid supplements among non-pregnant women ages 15–49 in Angul, Odisha, India, from linear regression equations & 95% confidence intervals (N = 3,793).**

|  | Model 1 | Model 2 |
|---|---|---|
| Village | -.00 (-.00-.00) | -.00* (-.00-.00) |
| Age | .00 (-.00 -.00) | -.01** (-.00-.00) |
| Education | .05* (.01-.09) | .02 (.00-.04) |
| Children | .02 (-.01-.05) | -.00 (-.02-.00) |
| Breastfeeding | .01 (-.00-.03) | .01* (.00-.01) |
| Descriptive norms | -.03 (-.07-.01) | -.02 (-.02-.01) |
| Injunctive norms | .36*** (.31-.41) | .11*** (.07-.13) |
| Collective norms | .25*** (.20-.30) | .07*** (.03-.08) |
| Knowledge | | .02* (.00-.04) |
| Self-efficacy | | .77*** (.71-.78) |
| Risk perception | | -.00 (-.02-.01) |
| Perceived access | | .00 (-.02 .02) |
| Outcome expectations | | .05*** (.06-.11) |
| (Adjusted R-squared) | (0.23) | (0.74) |

*Notes*: Iron folic acid supplement (IFA). Collective norms refer to collective norms around intentions to take iron folic acid supplements (IFA) and are calculated as the non-self mean of respondents within each village without the target person. All variables are standardized.

*p < .05

**p < .01

***p < .001.

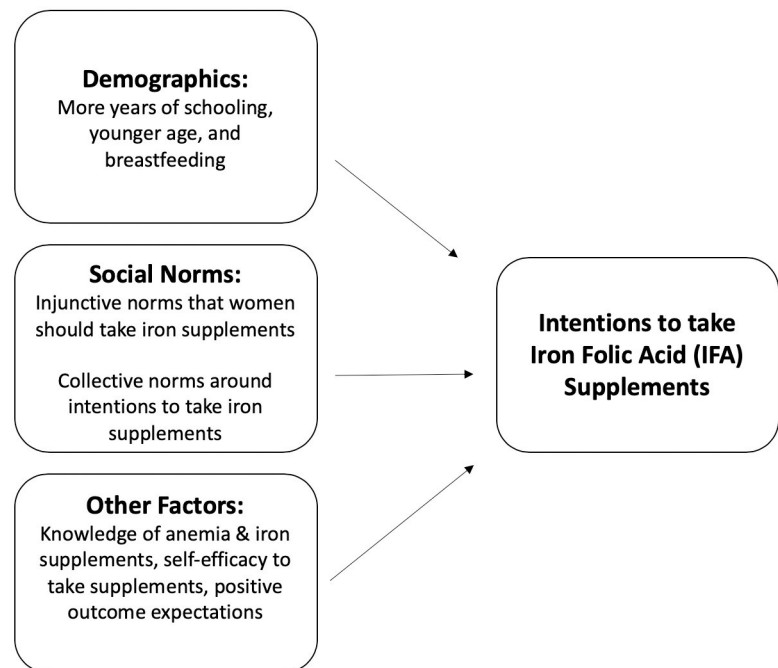

**Fig 1. Visual depiction of factors associated with intentions to take iron folic acid supplements.**

norms and collective norms diminished in model 2 when we added additional variables to the model but they remained significant. Including these factors in model 2 explained an additional 51% of the variance in intentions to take IFA (74% total variance). (see **Fig 1** for a visual depiction of the factors associated with intentions to take IFA).

Lastly, to understand which injunctive norms messages might have the greatest intervention impact, we also examined the association between each injunctive norm question separately and intentions to take IFA (rather than as an index). We found that they were all significantly associated with IFA intentions but the association was strongest when husbands expected their wives to take IFA, ($\beta = .08$, $p < .001$) compared to mothers-in-law ($\beta = .06$, $p < .001$), and other women in the community ($\beta = .06$, $p < .001$).

## Discussion

This study describes the current state of IFA behaviors among non-pregnant women in Odisha, India and identifies factors associated with intentions to take IFA supplements to inform a behavioral intervention to increase uptake. Despite Indian government guidelines that women of reproductive age should be taking weekly IFA regardless of anemia status, we found that less than 3% of the 3,914 non-pregnant women in our sample ($n = 113$), were currently taking them. We also found that women's perceptions about whether or not they had anemia was far from accurate. Only 5% of women self-reported that they were anemic, while according to hemocue testing measure, finger prick readings, 63% were actually anemic. It is important to note that 38% did not know whether or not they were anemic. Clearly, there is a knowledge gap and an intervention that conducts regular anemia testing could raise awareness about individual and community-wide anemia prevalence.

Given that so few women were actually taking IFA and because we wanted to inform a behavior change intervention, we examined associations with intentions to take iron supplements rather than IFA use itself. We found that while IFA use was low, intentions were rather

high. This could mean that despite high intentions, their behavior has not yet caught up, but they are primed for an IFA intervention. This knowledge provides valuable information around where to start communication efforts. In thinking about the Transtheoretical Model (also called the Stages of Change Model), developed by Prochaska and DiClemente in the late 1970's, that depicts the different stages needed to adopt and adhere to a behavior (readiness to change), women in this population are generally past the pre-contemplation phase and in the contemplation stage [34]. The forthcoming behavioral intervention will attempt to move bene-ficiaries to action (taking IFA) and subsequently to adherence. In practice this may mean that the intervention spends fewer resources on basic education about what IFA is as part of our communication strategy. Additionally, given that so few women knew that they were anemic but intentions to take IFA were moderately high, rather than spending the first few months of the intervention on anemia education, we will move directly into anemia testing. To take an individual behavior, anemia testing, and make it social, we will share community-level anemia test results. As women increase their IFA use over the course of the intervention and move into the "action stage" of the "stages of change" model, we will include adherence messaging that women should continue to take IFA throughout their whole reproductive lifecourse, the maintenance stage.

Our first model that focused on social norms showed that contrary to prior research, we did not find descriptive norms to have a significant impact on behavioral intentions [35,36]. This is not surprising given that descriptive norms were so low, $M = 0.25$, on a four-point scale with higher scores meaning perceptions that more non pregnant women were taking IFA. The variance was also low, $SD = 0.51$, meaning that this belief was consistent across the sample. In other words, women were accurately reporting that few others in their social networks were taking IFA, and this lack of variance could have suppressed its effects on intentions. It may also be the case that descriptive norms have an effect on actual IFA use (behavior) but not intentions to take IFA. Future research should test this relationship.

Another possibility around why descriptive norms were not associated with intentions may be that our measure was not robust enough. We assessed it with just a single measure (what proportion of non-pregnant women in your community take IFA), which likely introduced a significant level of error in the data.

It is worth noting that other research on the theory of normative social behavior has found that self-efficacy moderates the relationship between descriptive norms and behaviors [37], as do outcome expectations [38], issue familiarity [39], and external monitoring [40]. These findings, together with those reported in this paper, remind us that, to take advantage of the influence of social norms on behaviors, interventions may want to change social norms (e.g., increase per-ceived injunctive norms that non-pregnant women should be taking IFA) as well as these key moderators (that, in turn, augment the influence of norms). Raising awareness about anemia (to improve issue familiarity), highlighting the fact that taking IFA is easy (to enhance self-efficacy), that doing so is beneficial (outcome expectations), and asking husbands to ensure their wives are taking IFA (to enhance monitoring) are strategies that interventions could adopt.

In our sample, injunctive norms were higher on average than descriptive norms *(M* = 1.8 on a four point scale with higher scores meaning higher expectations that non pregnant women take IFA) and the deviation in descriptive social norms was larger, meaning that there was more variance in the responses (*SD* = .89). Injunctive norms were also significantly associated with intentions to take IFA. Given the importance of perceived injunctive norms from husbands in particular and also mother's-in-law, and community members, efforts to communicate national recommendations that non-pregnant should take IFA are likely to be more effective if they tar-get these audiences in addition to women themselves. Perhaps given our findings, injunctive norms, the belief that one *should* be taking IFA, is a logical place to start among non-pregnant

women. Of course, as IFA use and descriptive norms change throughout the course of an intervention, adding in descriptive norms messaging could be useful. Monitoring and evaluation data or midterm findings can help determine when IFA use increases and if/when descriptive norms change so that interventions can incorporate this messaging when appropriate.

Finally, collective norms, women living in a village in which others also have intentions to take IFA, were associated with individual intentions. Other studies have also found a similar effect of collective norms on individual attitude and behavior, in the domains of contraception use and attitudes about gender equity [21,32]. An intervention that not only takes an individual approach but one that communicates community level practices and beliefs may be more effective than one that ignores community influences.

Our findings that age, breastfeeding, knowledge, self-efficacy to take iron supplements, and outcome expectations around taking IFA were all significantly associated with intentions to take them have implications for future interventions. Since age is negatively associated with IFA intentions, targeting younger women of reproductive age who are still in their childbearing years (as IFA is currently only associated with pregnancy) could be a more promising approach. Additionally, knowledge only had a minor association with intentions to take IFA ($\beta = .02$, $p < 0.05$) and given that intentions to take IFA are already rather high, educating women about the basics of IFA and anemia will be included but not the focus of our communication efforts. Outcome expectations, the benefits of taking IFA to reduce anemia, had a slightly larger association with intentions to take IFA, ($\beta = .05$, $p < .001$), so intervention messaging will focus on the benefits of taking IFA to reduce anemia specifically for non-pregnant women. Prior research shows that some programs promoted IFA use but women did not understand why IFA was so important to their health [41]. Our intention variable did not state that women intended to take IFA throughout their reproductive years. If we are expecting women to take weekly IFA throughout their reproductive life cycle (ages 15–49), the national and global recommendation, then women need to understand the benefits.

Perhaps one of our most significant findings was the strength of the association between self-efficacy beliefs and intentions. In the multivariable model, the standardized beta coefficient of this relationship was high, ($\beta = .77$, $p < .001$), signifying its robust association with intentions, even after controlling for other factors in the model. Our finding further illustrates the importance of self-efficacy that has been documented in numerous other studies [42,43]. Improving one's self-efficacy to take IFA by designing interventions that address barriers that women report, is a logical first step to increase intentions and ultimately IFA use. For example, in our formative research, we found that common barriers include lack of knowledge that non-pregnant women should be taking IFA, inequitable gender norms that women focus on the family's well-being rather than her own, and simply remembering to take it [14,44,45]. To address these barriers and to improve self-efficacy, our intervention will focus on raising awareness that non-pregnant women should be taking IFA and it is expected that they take it (injunctive norms) via short videos where non-pregnant women like them are taking it and their husbands and mothers-in-law encourage them to prioritize their own health. To address forgetfulness to take IFA, we will consider distributing a calendar to post in participants homes to remind them when to take IFA, which family and others who visit their home, can see to help change norms that non-pregnant women should be taking it and help increase one's self-efficacy to remember to take it.

## Limitations

Our study has some limitations that may affect the interpretation of the results. Despite literature that points to supply side factors, we did not account for supply side barriers. We also did

not account for frontline health worker attitudes about distributing IFA to non-pregnant women or local policy related barriers or facilitators. These environmental factors may be particularly important barriers to actual use. Accounting for these factors may elucidate the additional missing variance in our model. However, our model did account for 74% of variance in IFA intentions. Furthermore, intention to consume IFA is used as the primary outcome in this study. While intentions have been found to be a significant predictor of behaviors, other structural barriers related to the supply and distribution of IFA may prevent intentions from actually being translated into action [46,47]. Also, the meta-analysis conducted by Cooke et al., 2016 found that intention is a better predictor for older populations, as younger populations are less likely to translate their intentions into behavior. Nevertheless, intention was significantly associated with behaviors in our sample. We also chose to operationalize intentions based on barriers that we found in our formative research (e.g., taking it even if you are not pregnant and taking it even if your husband or mother-in-law does not want you to). While these questions illuminated specific sides of intentions, a simple question that did not take into account these barriers may have produced different results.

## Conclusion

These findings can help program planners and researchers designing an anemia reduction intervention decide when and how to invest resources to change intentions to take IFA, thereby increasing use and reducing anemia prevalence in rural India. While these findings are context specific, other iron supplement or anemia reduction interventions that do not have the luxury of doing a full formative research assessment or a comprehensive baseline assessment, could use these findings as a starting point.

Overall, our results indicate that shifting social norms within communities and improving people's knowledge are not enough to drive behavior if individuals do not believe that they can engage in that behavior (self-efficacy) and that the behavior will result in desired outcomes (outcome expectations). Our results also show that in populations where the behavior of interest is extremely low, focusing on injunctive norms (expectations) and potentially adding descriptive norms messaging (that women are in fact enacting the behavior) after behaviors have changed, is a logical order or operations. To do this, monitoring and evaluation or mid-term data is necessary to determine that behaviors have indeed changed and interventionists can accurately state that many women or a specific percentage of women in their community are enacting the behavior.

While the Indian government committed to reducing anemia among non-pregnant women of reproductive in 2013, in this sample of almost 4,000 women, less than 3% of women are actually taking IFA. If the World Health Organization wants to meet their global target of a 50% reduction in anemia by 2025, they need to change course fast [2]. While women who are not pregnant make up the largest sub group of people with anemia, compared to children and pregnant women, findings from this study show that they are largely disregarded. Specific targeted efforts towards women at different stages of their reproductive life course (adolescents, non-pregnant women, and pregnant women) is a logical first step to increase iron supplement use, and thereby reduce anemia.

## Supporting information

**S1 Appendix.**
(DOCX)

## Author Contributions

**Conceptualization:** Erica Sedlander, Michael W. Long, Jeffrey B. Bingenheimer, Rajiv N. Rimal.

**Data curation:** Erica Sedlander.

**Formal analysis:** Erica Sedlander.

**Funding acquisition:** Rajiv N. Rimal.

**Investigation:** Erica Sedlander.

**Methodology:** Erica Sedlander, Michael W. Long, Jeffrey B. Bingenheimer, Rajiv N. Rimal.

**Project administration:** Erica Sedlander.

**Software:** Erica Sedlander.

**Supervision:** Jeffrey B. Bingenheimer, Rajiv N. Rimal.

**Writing – original draft:** Erica Sedlander.

**Writing – review & editing:** Erica Sedlander, Michael W. Long, Jeffrey B. Bingenheimer, Rajiv N. Rimal.

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
