## [Decision Letter · Decision Letter 0]

23 Dec 2020

PONE-D-20-27062

Examining intentions to take iron supplements to inform a behavioral intervention: The Reduction in Anemia through Normative Innovations (RANI) Project

PLOS ONE

Dear Dr. Long,

Thank you for submitting your manuscript to PLOS ONE. After careful consideration, we feel that it has merit but does not fully meet PLOS ONE’s publication criteria as it currently stands. Therefore, we invite you to submit a revised version of the manuscript that addresses the points raised during the review process.

Two reviewers with expertise in behaviour change and research in social norm have independently reviewed your manuscript and recommended minor revision. Indeed, your manuscript is well-written and addresses an important topic that is relevant to both researchers and practitioners. Please read carefully the reviewers' comments and provide point-by-point responses (or rebuttal) in your response letter. However, please not that revising your manuscript does not guarantee publication to PLOS One. The reviewers will be asked to re-review the revised manuscript and, if the issues they have raised are satisfied, they will provide their recommendation for publication. 

We look forward to receiving your revised manuscript.

Kind regards,

Lambros Lazuras

Academic Editor

PLOS ONE

Journal Requirements:

Reviewers' comments:

Reviewer's Responses to Questions

**Comments to the Author**

1. Is the manuscript technically sound, and do the data support the conclusions?

Reviewer #1: Yes

Reviewer #2: Yes

2. Has the statistical analysis been performed appropriately and rigorously? 

Reviewer #1: Yes

Reviewer #2: Yes

3. Have the authors made all data underlying the findings in their manuscript fully available?

Reviewer #1: Yes

Reviewer #2: Yes

4. Is the manuscript presented in an intelligible fashion and written in standard English?

Reviewer #1: Yes

Reviewer #2: Yes

5. Review Comments to the Author

Reviewer #1: See attached file xxxxxxxxxxxxxxxxxxxxxxxxxxxxxxxxxxxxxxxxxxxxxxxxxxxxxxxxxxxxxxxxxxxxxxxxxxxxxxxxxxxxxxxxxxxxxxxxxxxxxxxxxxxxxxxxxxxxxxxxxxxxxxxxxxxxxxxxxxxxxxxxxxxxxxxxxxxxxxxxxxxxxxxxxxxxxxxxxxxxxxxxx

Reviewer #2: Dear authors,

Congratulations to a very professionally written, highly relevant and fascinating study on anaemia in non-pregnant women in India.

I have a few minor comments and suggestions to improve your manuscript.

1. In general: Worth mentioning why the rate of anaemia in women In India is so high? Is it unusually high compared to other countries? If yes, why?

2. Page numbers are always p10

3. L 14. Maybe revise sentence. It still sounds like pregnant women have more of a right not to be anaemic compared to non-pregnant women. I would recommend a formulation like “Both, pregnant and noon-pregnant women are equally…”

4. L 33 guidelines stating that…

5. L 34 ‘evidence’ to practice gap? Is this the right term here? I understand the problem is that the government does not practice what they preach? Not sure if evidence is the correct word

6. L 47-53 explain why norms are the main focus as compared to the other variables

7. L 54-56 very nice part, gap is explained very well

8. L 74 maybe add Schultz 2007 reference on inj and desc norms

9. L 91 IFA intentions? Rethink terminology. IFA-taking intentions? Also, stick to one terminology for intentions throughout your manuscript. They are changing slightly throughout.

10. L 102 onwards. Check numbers and written-out numbers. Sometimes you say 22%, sometimes twenty two percent.

11. L 134 term “data collector”. Local enumerators? Research Assistants? Mention if they received previous training as well please

12. L 145 Why was mobile phone ownership added as demographic variable?

13. L 161 Did you look at individual inj norms? Was there one norm that was especially strong? E.g. the husband inj norm explained most variance? That would be interesting as well for the communication aspect

14. L 197 You talk about during pregnancy suddenly whilst the whole study is about non-pregnant women?

15. L 205 well done pre-testing your items. Why did they only perceive this difficulty in this item? Was the Likert scale appropriate for all other items?

16. L 211 I have difficulties with the way you operationalized intentions. Why is there no basic intention to take IFA? Each intentions has a second part to it that says “even if” which is a mix of norms, self-efficacy and intentions. I think this needs to be discussed in the limitations and explained why this was needed. In my eyes, you cover self-efficacy and norms very well already and a basic three-item measure of intentions would suffice. Of course you can’t change this in retrospect but maybe discuss shortly and explain why this kind of intentions.

17. L 290-301 onwards: maybe a bit more detail about what the stage model means and what kind of intervention fits your identified stage (inspiration from Bamberg 2013)

18. L 302-310 it could also mean that actually, Descriptive norms have a lot of power influencing actual behaviour (as compared to intentions, because not many take IFA) but that it is not an explicit influence of norms as demonstrated in Nolan, Schultz et al 2008.

19. L321 what does improve social norms mean?

20. L 328 deviation in descriptive social norms was larger…

21. L 340-346 I really like this paragraph.

22. L 353 increasing knowledge contradicts your Stage model claim?

23. I miss more specific ideas in the discussion about how to operationalize an intervention, what has been tested before to increase self-efficacy in a similar case for example and how would you do it?

24. L 369 Limitations. Did all people find it easy answering the scales? I find it sometimes hard to conduct survey studies with people from various background, which means facilitators have to engage more or less with some people which could lead to biases?

25. L 392 – 395 very nice section

26. References. Make sure you add DOIs

I wish you the best of luck and merry Christmas.

6. PLOS authors have the option to publish the peer review history of their article (what does this mean?). If published, this will include your full peer review and any attached files.

Reviewer #1: **Yes: **Sarah Cotterill

Reviewer #2: No

---

## [Author Response · Author response to Decision Letter 0]

22 Jan 2021

We have included responses to all reviewer comments in the attached Response to Reviewers table. Please let us know if this requires any changes.

---

## [Decision Letter · Decision Letter 1]

3 Mar 2021

PONE-D-20-27062R1

Examining intentions to take iron supplements to inform a behavioral intervention: The Reduction in Anemia through Normative Innovations (RANI) Project

PLOS ONE

Dear Dr. Long,

Thank you for submitting your manuscript to PLOS ONE. I am happy to accept your manuscript for publication subject to some very minor comments addressed by both reviewers. Therefore, we invite you to submit a revised version of the manuscript that addresses the points raised during the review process.

We look forward to receiving your revised manuscript.

Kind regards,

Lambros Lazuras

Academic Editor

PLOS ONE

Journal Requirements:

Reviewers' comments:

Reviewer's Responses to Questions

**Comments to the Author**

1. If the authors have adequately addressed your comments raised in a previous round of review and you feel that this manuscript is now acceptable for publication, you may indicate that here to bypass the “Comments to the Author” section, enter your conflict of interest statement in the “Confidential to Editor” section, and submit your "Accept" recommendation.

Reviewer #1: (No Response)

Reviewer #2: All comments have been addressed

2. Is the manuscript technically sound, and do the data support the conclusions?

Reviewer #1: Yes

Reviewer #2: (No Response)

3. Has the statistical analysis been performed appropriately and rigorously? 

Reviewer #1: Yes

Reviewer #2: Yes

4. Have the authors made all data underlying the findings in their manuscript fully available?

Reviewer #1: Yes

Reviewer #2: Yes

5. Is the manuscript presented in an intelligible fashion and written in standard English?

Reviewer #1: Yes

Reviewer #2: Yes

6. Review Comments to the Author

Reviewer #1: The authors have fully revised the paper in response to the comments from reviewers. I am satisfied that my comments have been fully addressed, except for one small point on which I would like to see more detail:

In response to my previous comments the authors have amended the Methods section to make it clear that stratification ‘by the proportion of minority populations’ and 20 clusters were selected from each strata. It would be informative to specify how the strata were categorised, for example, were the proportions of minority populations stratified into x groups according to some cut-points and then y clusters chosen from each strata? (page 9, line 22).

Other than this small issue, this is a well-written paper on an interesting topic and I recommend it be accepted for publication.

Reviewer #2: Dear authors,

Thank you for addressing the reviewer comments so thoroughly.

I am adding a few more minor comments that can be addressed:

1. Abstract background: replace “ignored” with “disregarded” (sounds less intentional)

2. Abstract methods spell out “other factors” (name these factors)

3. Abstract results: you cross out education, did you exclude it now? Did you include it before? Mention in the paper that you measured but excluded it from the analysis then.

4. Abstract conclusion: “logical first step” replace with “the step we recommend based on our results”

5. L 5: change order of sentence “a 50% reduction… is the second global target by the WHA”

6. L 16: replace deserve with shall be supported

7. L 22: poverty is often related to education as well, so definitely needs to be explained why you took it out of the model (would be a relevant predictor). Is Malaysia and infectious diseases related to iron deficiency as well?

8. L 29 change location of “weekly” back to where it was

9. L 36 ff: Divide into 2 sentences to make clear

10. L 52: delete “to use”

11. P 6 and 7 very nice changes and additions

12. L 112 reformulate “at different times” over time? Prone to change?

13. L 152-164 very nicely explained

14. L 178 consent for what?

15. L 296: You indicate education level here? Did you exclude it or not?

16. L 300 ff and 381 ff: thought: is there any information or research about father’s husbands attitude to iron supplements? It seems important for women that the supplements are approved by their male household members, so maybe the campaign needs to target them in the first place? Do we know how they think about it? Do they support their daughters/wives or are they sceptical? I recommend going a little bit deeper into this aspect, also in the introduction and the implications in the end.

17. The discussion improved a lot.

18. L 521 ff: lovely finish

7. PLOS authors have the option to publish the peer review history of their article (what does this mean?). If published, this will include your full peer review and any attached files.

Reviewer #1: **Yes: **Dr Sarah Cotterill

Reviewer #2: **Yes: **Isabel Richter

---

## [Author Response · Author response to Decision Letter 1]

16 Mar 2021

We have included a response to reviewer table addressing all reviewer comments.

---

## [Editor Report · Decision Letter 2]

23 Mar 2021

Examining intentions to take iron supplements to inform a behavioral intervention: The Reduction in Anemia through Normative Innovations (RANI) Project

PONE-D-20-27062R2

Dear Dr. Long,

We’re pleased to inform you that your manuscript has been judged scientifically suitable for publication and will be formally accepted for publication once it meets all outstanding technical requirements.

Kind regards,

Lambros Lazuras

Academic Editor

PLOS ONE
---

## [Editor Report · Acceptance letter]

29 Apr 2021

PONE-D-20-27062R2 

Examining intentions to take iron supplements to inform a behavioral intervention: The Reduction in Anemia through Normative Innovations (RANI) Project 

Dear Dr. Long:

I'm pleased to inform you that your manuscript has been deemed suitable for publication in PLOS ONE. Congratulations! Your manuscript is now with our production department. 

Kind regards, 

on behalf of

Dr. Lambros Lazuras 

Academic Editor

PLOS ONE